# First Description of the Nuclear and Mitochondrial Genomes and Associated Host Preference of *Trichopoda pennipes*, a Parasitoid of *Nezara viridula*

**DOI:** 10.3390/genes14061172

**Published:** 2023-05-27

**Authors:** Mesfin Bogale, Shova Mishra, Kendall Stacey, Lillie Rooney, Paula Barreto, Gina Bishop, Katherine Bossert, Kalista Bremer, Daniel Bustamante, Lila Chan, Quan Chau, Julian Cordo, Alyssa Diaz, Jordan Hacker, Lily Hadaegh, Taryn Hibshman, Kimberly Lastra, Fion Lee, Alexandra Mattia, Bao Nguyen, Gretchen Overton, Victoria Reis, Daniel Rhodes, Emily Roeder, Muhamed Rush, Oscar Salichs, Mateo Seslija, Nicholas Stylianou, Vivek Vemugunta, Min Yun, Anthony Auletta, Norman Leppla, Peter DiGennaro

**Affiliations:** Entomology and Nematology Department, University of Florida, Gainesville, FL 32611, USA; azene.2@osu.edu (M.B.); shovamishra@ufl.edu (S.M.); kstacey@ufl.edu (K.S.); rooney.lillie@ufl.edu (L.R.); pbarreto@ufl.edu (P.B.); gina.bishop@ufl.edu (G.B.); kbossert@ufl.edu (K.B.); kalista.bremer@ufl.edu (K.B.); danielbustamante@ufl.edu (D.B.); lilachan@ufl.edu (L.C.); qchau@ufl.edu (Q.C.); juliancordo@ufl.edu (J.C.); diaz.alyssa@ufl.edu (A.D.); jordan.hacker@ufl.edu (J.H.); lilyhadaegh@ufl.edu (L.H.); taryn.hibshman@ufl.edu (T.H.); kimberlylastra@ufl.edu (K.L.); f.lee@ufl.edu (F.L.); amattia@ufl.edu (A.M.); nguyen.bao@ufl.edu (B.N.); overton.gretchen@ufl.edu (G.O.); vreis1@ufl.edu (V.R.); danielrhodes@ufl.edu (D.R.); emilyroeder@ufl.edu (E.R.); mrush1@ufl.edu (M.R.); oscarsalichs@ufl.edu (O.S.); mateoseslija@ufl.edu (M.S.); nstylianou@ufl.edu (N.S.); vvemugunta@ufl.edu (V.V.); yun.min@ufl.edu (M.Y.); anthonyauletta@ufl.edu (A.A.); ncleppla@ufl.edu (N.L.)

**Keywords:** *Trichopoda pennipes*, *Nezara viridula*, *Leptoglossus phyllopus*, mitogenome, genome, sibling species

## Abstract

*Trichopoda pennipes* is a tachinid parasitoid of several significant heteropteran agricultural pests, including the southern green stink bug, *Nezara viridula*, and leaf-footed bug, *Leptoglossus phyllopus*. To be used successfully as a biological control agent, the fly must selectively parasitize the target host species. Differences in the host preference of *T. pennipes* were assessed by assembling the nuclear and mitochondrial genomes of 38 flies reared from field-collected *N. viridula* and *L. phyllopus*. High-quality de novo draft genomes of *T. pennipes* were assembled using long-read sequencing. The assembly totaled 672 MB distributed among 561 contigs, having an N50 of 11.9 MB and a GC of 31.7%, with the longest contig at 28 MB. The genome was assessed for completeness using BUSCO in the Insecta dataset, resulting in a score of 99.4%, and 97.4% of the genes were single copy-loci. The mitochondrial genomes of the 38 *T. pennipes* flies were sequenced and compared to identify possible host-determined sibling species. The assembled circular genomes ranged from 15,345 bp to 16,390 bp and encode 22 tRNAs, two rRNAs, and 13 protein-coding genes (PCGs). There were no differences in the architecture of these genomes. Phylogenetic analyses using sequence information from 13 PCGs and the two rRNAs individually or as a combined dataset resolved the parasitoids into two distinct lineages: *T. pennipes* that parasitized both *N. viridula* and *L. phyllopus*, and others that parasitized only *L. phyllopus.*

## 1. Introduction

The invasive southern green stink bug, *N. viridula* (Hemiptera: Pentatomidae), continues to spread throughout the southern US and into the northeastern states [1]. Being highly vagile, it has become a cosmopolitan pest [2]. It is polyphagous, infesting many economically important crops including grains, fruits, nuts, and vegetables, with a strong preference for legumes [3]. *N. viridula* is among the predominant species of stink bugs that reduce soybean yields in the South each year [4] and the pest will continue to increase in importance as worldwide soybean production expands. Chemical insecticide applications often are ineffective against the southern green stink bug [5]. Current attempts to manage pest stink bugs rely primarily on broad-spectrum insecticides, such as organophosphates, pyrethroids, and neonicotinoids because selective chemistries are unavailable [1,6]. Broad-spectrum insecticides, however, negatively impact natural enemies [6,7] and disrupt management tactics for other agricultural pests. Organic growers have fewer and less effective options for controlling stink bugs [8].

There is a recognized need and considerable potential for inoculative and augmentative biological control programs to reduce the impact of southern green stink bugs on crops [9,10,11]. In North America, *T. pennipes* Fabr. (Diptera: Tachinidae) (Figure 1) is the primary parasitoid of the adult, the most difficult life stage to control [12]. *N. viridula* has been managed with biological control completely or at least partially in Hawaii, Australia, and New Zealand using imported *T. pennipes* and hymenopteran egg parasitoids [13,14,15]. Biological control programs generally are more effective when multiple parasitoid species that attack different life stages of the target pest are used in concert. However, *T. pennipes* failed to establish on *N. viridula* in Australia, New Zealand, Fiji, New Guinea, the Solomon Islands, and South Africa [16]. Conversely, *T. pennipes* from an undetermined location appeared in central Italy in 1988 and subsequently parasitized only *N. viridula* [17]. Indigenous *T. pennipes* in California do not parasitize the squash bug, *Anasa tristis* (DeGeer) (Hemiptera: Coreidae), but this pest eventually was managed by importing, rearing, and releasing *T. pennipes* from states in the northeastern US [11,18]. It is essential to understand what caused some of these biological control attempts with *T. pennipes* to succeed or fail [19] and the scientific basis for this potentially irreversible action [20].

*Trichopoda pennipes* parasitizes 4th and 5th instar nymphs and adults of the southern green stink bug [2,21]. It also parasitizes the leaf-footed bug, *L. phyllopus* (L.) (Hemiptera: Coreidae), and other indigenous coreids, such as *A. tristis*, its principal host in the northcentral and northeastern US [21,22]. *T. pennipes* fortuitously also has begun to parasitize the brown marmorated stink bug, *Halyomorpha halys* (Stål), an adventive pentatomid discovered in Pennsylvania following its invasion of the US from Asia in the 1990s [12,23]. In the southern US, *N. viridula* is the primary host of *T. pennipes* [24], although the parasitoid readily oviposits on *L. phyllopus*. *T. pennipes* indigenous to California oviposits on the bordered plant bug, *Euryopthalmus cinctus californicus* Van Duzee (Hemiptera: Largidae), but not on *A. tristis*. [11]. Consequently, host-determined sibling species are assumed to exist [11,12,18]. Molecular analysis is needed to determine if *T. pennipes* reared and released as a biological control agent will parasitize the target host species in a specified area [25].

Genomic information has not been reported for *T. pennipes*. Therefore, we sequenced and assembled the nuclear and mitochondrial genomes of flies derived from parasitized *N. viridula* and *L. phyllopus* to determine if mitochondrial genotypic differences existed and could be characterized. The purpose was to establish methods for identifying and rearing *T. pennipes* that prefer targeted species of heteropteran agricultural pests. This research provides genomic resources for elucidating the host preference of this parasitoid and foundational molecular diagnostic methods for developing *T. pennipes* lines for laboratory colonization.

## 2. Materials and Methods

### 2.1. Sample Collection and Identification

Parasitized adult *N. viridula* and *L. phyllopus* were hand-collected from a grain sorghum crop at the UF/IFAS North Florida Research and Education Center, Suwannee Valley near Live Oak, Florida from July to September 2021. The insects were identified using a key to the “Hemiptera of US Authors” (J. E. Eger, unpublished), and the two host species were kept in separate 23 cm × 23 cm × 70 cm BugDorms (MegaView Science Co., Ltd., Taichung, Taiwan) in a walk-in environmental chamber at 25 °C, 66% RH, and a 14L:10D photoperiod. Each BugDorm was provisioned with a 60-cc plastic water cup with a 10-cm long dental wick stuck through a hole in the lid. The adults were given 5 cm lengths of fresh organic green beans and raw shelled peanuts. The bottom of each BugDorm was checked daily for the emergence of *T. pennipes* larvae and subsequent pupae. The pupae from *N. viridula* and *L. phyllopus* were removed and placed in separate 2-oz. plastic solo cups with a maximum of ten pupae each. Each cup had a small ventilation hole in the lid, and a shallow layer of moist sterilized soil to support the pupae until the emergence of adults for DNA extraction. For mitogenome sequencing, DNA was extracted using Quick-DNA Miniprep Plus Kits (Genesee Scientific, El Cajon, CA, USA) according to the manufacturer’s instructions from a total of 38 *T. pennipes* samples; 27 from parasitoids of *N. viridula* and 11 from parasitoids of *L. phyllopus* and labeled NV and LP, respectively, based on the source-host. For nuclear genome sequencing, DNA was extracted from one *T. pennipes* sample derived from a parasitized *N. viridula* (NV2).

### 2.2. PCR Amplification and Sequencing

For whole genome sequencing, the DNA sample from NV2 was sent for HiFi PacBio sequencing at the University of Florida Interdisciplinary Center for Biotechnology Research (ICBR) NextGen DNA Sequencing core facility. ICBR performed library preparation and sequencing was done on a single lane.

For mitochondrial genome sequencing, the mitochondrial genome was amplified for each of the 38 parasitoid samples using a set of primers (FY-J-214, FY-N-1873, FY-J-2198, FY-N-3705, FY-J-4463, FY-N-5748, FY-J-5747, FY-N-6160, FY-J-7077, FY-N-7793, FY-J-7572, FY-N-8741, FY-J-8641, FY-N-9629, FY-N-13889, and FY-14722) [26], employing one shot LA PCR^TM^ Mix (Ver. 2.0, TaKaRa Bio Inc., San Jose, CA, USA). Each PCR (50 µL) contained 25 µL PCR mix, 0.2 µM of each primer, and 0.4 µM of DNA. The PCR cycle consisted of an initial denaturation at 95 °C for 4 min followed by a five-cycle touchdown phase starting at 60 °C and decreasing by 1 °C/cycle. This was followed by 30 cycles of denaturation at 95 °C for 45 s, annealing at 55 °C for 30 s, and extension at 72 °C for 2 min. The reaction was terminated after a final extension at 72 °C for 10 min. PCR was performed on a GeneAmp PCR System 2700 (Applied Biosystems, Thermo Fisher Scientific, Waltham, MA, USA). Amplification products were cleaned using a Zymoclean Gel DNA Recovery Kit (Genesee Scientific, USA). Samples were barcoded and sequenced in a single lane.

### 2.3. Whole Genome Sequence Analysis

PacBio HiFi raw reads were first checked for quality using FastQC [27]. Clean reads were subjected to de novo assembly using the HiFiasm assembler [28] with an aggressive duplicate purging parameter. Assembled genome was checked for contamination. To remove any possible bacterial contamination, Blastn was used to conduct a sequence search of all the contigs against the NCBI nucleotide database (with E-value cutoff < 1 × 10^−5^), and taxonomy was assigned to each contig. Each raw read was mapped to contigs using Bowtie2 [29]. Then, any possible contamination was determined and visualized using Blobtools v.1.0 [30]. The quality of the draft genome was determined by Quast [31], and its completeness was assessed using BUSCO v.5.2.0 [32].

### 2.4. Mitogenome Annotation and Sequence Analysis

Raw reads were assembled for each sample using the invertebrate mitogenome genetic code in MitoFinder v.1.4. [33] with *Exophasia rotundiventris* (GenBank No: NC_050938.1) as a reference genome. Assembled complete genomes were analyzed for genome architecture using SnapGene (Insightful Science; Snapgene.com accessed on 26 January 2022). Next, sequences of 13 PCGs, and the small and large subunit rRNA genes (srRNA and lrRNA, respectively), were extracted for each sample employing MitoFinder. Twenty-three samples containing high-quality sequences of 15 genes were analyzed for their phylogenetic relationships. Nucleotide sequences were aligned using Clustal Omega [34], and the alignments were manually corrected in PAUP (Version 4.0b; [35]) where needed. Aligned sequences were trimmed at either end to match the sizes of some gene sequences for which we only had partial sequence information. Maximum likelihood (ML) and maximum parsimony (MP) trees were generated using individual and combined sequence information from these genes in MEGA-X [36,37] using *E. rotundiventris* (GenBank No: NC_050938.1) as an outgroup. For ML analyses, initial tree(s) for heuristic searches were obtained automatically by applying Neighbor-Join and BioNJ algorithms to a matrix of pairwise distances estimated using the maximum composite likelihood approach and then selecting the topology with superior log likelihood values. For MP analyses, heuristic searches were based on tree bisection-reconnection. Bootstrap analyses [38] were based on 1000 replications.

## 3. Results

### 3.1. Nuclear Genome Sequence Analyses

A total of 3.3 million HiFi PacBio reads were generated with an average read length of 5647 bp and average read quality of 39. All high-quality reads were assembled de novo using HiFiasm. The bacterial contamination was less than 0.3% of total sequences (Appendix A). The genome size of *T. pennipes* was 672 MB distributed among 561 contigs, with the largest contig at 28 MB. The GC content was 31.7%. and the N50 value is 11.9 MB (Table 1). A total of 1367 BUSCO in the Insecta dataset were used, and the draft genome of *T. pennipes* had a complete BUSCO score of 99.4%. Most of these genes were single-copy loci with 0.2% fragmented and 0.4% missing BUSCOs (BioProject PRJNA971909).

### 3.2. Mitochondrial Genome Sequencing and Assembly

A total of 1,716,087 demultiplexed HiFi PacBio reads were obtained for the 38 *T. pennipes* samples, with an average read quality of 54.8 and length of 1117 bp (Table 2). All high-quality reads were used to assemble individual mitochondrial genomes (GeneBank accessions OR032586-OR032599). Complete *T. pennipes* mitochondrial genomes could be assembled for nine samples, four from parasitoids of *N. viridula* and five from parasitoids of *L. phyllopus*. For 14 isolates, we only obtained partial sequence information for some genes; sequence information from PCGs of these isolates have, however, been included in our phylogenetic analyses (see below). The sizes of complete mitogenomes for these parasitoids ranged from 15,345 bp to 16,390 bp, with the AT content ranging from 78.52% to 80.40%. The genome of NV2 had 15,396 bp (Figure 2) and 15 genes were analyzed (Table 3). All complete and partial assemblies encoded for 37 genes, including 13 PCGs, 22 tRNA genes, and two rRNA genes.

The total length of the 13 PCGs was 11,202 bp, with an average AT content of 79.30%. The lsRNA with 1291–1295 bp was located between tRNA-Leu and tRNA-Val, whereas the srRNA with 789 bp (except in NV25, 783 bp) was located between tRNA-Val and the control region as in other Tachinidae [39]. The average AT content of the rRNA was 82%. There were, however, significant differences in the AT-rich region located between the srRNA and tRNA both in terms of size and nucleotide composition, so it was impossible to align the nine complete genomes in this region.

### 3.3. Phylogenetic Analyses of Mitogenomes

Of the 23 samples constituting the ingroup, complete sequence information was obtained for eight of the 15 genes (13 PCGs plus two rRNA genes) (Table 3). Only partial sequence information could be retrieved for the remaining ten genes for some samples for lack of sufficiently good quality reads. This forced trimming of the ends of more complete gene sequences to match the incomplete sequence information. The 23-sample combined dataset contained 12,580 characters (including alignment gaps), of which 115 were phylogenetically informative. Maximum likelihood (ML) and Maximum parsimony (MP) analyses using combined sequence information from the 15 genes produced by phylogenic trees (Figure 3 and Figure 4). Both ML and MP analyses resolved the ingroup into Lineages 1 and 2; the former consisting of *T. pennipes* from both *N. viridula* (NV) and *L. phyllopus* (LP), and the latter exclusively populated by parasitoids from *L. phyllopus* (LP).

Nucleotide sequence information from each of the 15 mitochondrial genes analyzed individually could only separate Lineage 2 (the LP clade) as distinct from the rest of the ingroup; there was no bootstrap support for Lineage 1 (the LP/NV clade) in trees generated using sequence information from the individual genes (data not shown). The only exception was sequence information from Cox III, where these clades were resolved as distinct (Figure 5 and Figure 6). All phylogenies based on the combined sequence information or Cox III alone resulted in trees that had similar topologies. The main difference among these phylogenies was the number of bootstrap-supported branches in Lineage 1, and the level of bootstrap support for branches in both lineages.

## 4. Discussion

The goal of this research was to generate genome resources for *T. pennipes* and use them to determine if there are mitochondrial genotypic differences between parasitoids that emerged from *N. viridula* or *L. phyllopus*. The fly oviposits on both hosts despite the smaller *L. phyllopus* probably being less suitable for larval development. *T. pennipes* evolved with *L. phyllopus* and other coreids in the Nearctic and Neotropical regions, whereas *N. viridula* is a more recent association for the parasitoid, having originated in the Ethiopian region and entered North America sometime before 1880 [40]. Thus, one lineage of the parasitoid appears to have adapted to parasitizing the larger pentatomid host. Flies in the other lineage oviposit on the apparently less suitable host even when the more suitable host is present in the same area. In addition to size, the separation of the lineages could be mediated by different volatile compounds released from the two hosts acting as kairomones that attract flies from one or both lineages [41]. Correspondingly, two distinct lineages of *T. pennipes* were defined by the mitochondrial sequences, indicating the possible existence of two sibling species: one that parasitizes only the coreid, *L. phyllopus*, and the other that parasitizes both the coreid and pentatomid, *N. viridula.* Developing the tools and resources required to understand host preference in *T. pennipes* is essential for maximizing its use as a biological control agent.

We sequenced and assembled the genome of *T. pennipes* for the first time using HiFi PacBio sequencing. The 672 MB genome comprised 561 contigs where all contigs were larger than 6 KB. This genome size is consistent with data from other Diptera [42], although a flow cytometer could have been used to estimate the genome size more accurately. The contigs were blasted into five taxon groups and visualized in Blobplot. The contamination by fungi and bacteria was less than 0.3% of the total sequences, suggesting that the assembled *T. pennipes* genome is accurate. In its current state, this genome provided the basis for matching the parasitoid to its target host species.

There was no significant difference in mitochondrial genome architecture among the parasitoid flies that were sequenced. The entire and partial mitochondrial genome assemblies encoded 37 genes including 13 PCGs, 22 tRNA genes, two rRNA genes, and a non-coding AT-rich control region. The assemblies were all on circular mitochondrial DNA, typical of metazoan mitogenomes [43]. There also was no difference among assemblies in the synteny of these genes. For the 13 PCGs and two rRNA genes, only ND6, ND4L, and ATP6 were located on the minor (N) strand. Nine of these 15 genes (Cyt B, ND6, ND3, Cox III, ATP6, ATP8, Cox II, Cox I, and ND2) pointed in the same direction. In contrast, the remaining six were oriented in the opposite direction, which is assumed to be typical of the common ancestor of insects [39,44]. Further, based on the nine completed assemblies, there was no difference among almost all the isolates in terms of sizes of PCGs, tRNAs, and rRNAs. The only exceptions were lrRNA of NV4 which was three bp larger and NV25 one bp smaller; and srRNA of NV25 was smaller by six bp. There currently is no sequence information available on *T. pennipes* in the public databases. Therefore, the mitochondrial and nuclear genome sequence information generated in this study will be a significant contribution for this species and it will also support the comparative genomics of Diptera.

We conducted phylogenetic analyses of the *T. pennipes* using the 13 PCGs and two rRNAs individually or as a 15-gene combined dataset. All phylogenies based on the combined and individual Cox III datasets resolved the ingroup into two lineages; Lineage 1, consisting of parasitoids from both *N. viridula* and *L. phyllopus*, and Lineage 2, exclusively populated by parasitoids from *L. phyllopus*. Phylogenies based on the remaining 14 individual genes also resolved Lineage 2 as distinct. Separating the *T. pennipes* into two lineages is useful for assessing their host preference, which may impact their effectiveness in biological control. One explanation for the *T. pennipes* separating into different lineages may be that multiple siblings or cryptic species of the parasitoid coexist at our collection site. Three sibling species of *T. pennipes* have been proposed to exist in the United States: one in the Northeast, another in the Southwest, and a third in the Southeast [18]. These geographic populations were considered distinct sibling species because flies from the northeast and southwest would not mate in the laboratory. Additionally, the putative sibling species show differences in host preference. Flies in the Northeast mainly parasitize the squash bug, *A. tristis*, plus several other coreids and some pentatomids. In the Southwest, indigenous *T. pennipes* only parasitize *E. cinctus californicus.* The Southeastern flies parasitize coreids, including *A. tristis* and *L. phyllopus*, along with *N. viridula* and several other pentatomids [2].

The hypothesis that sibling species of *T. pennipes* coexist in Florida is supported by the sequence data that revealed separate mitochondrial markers in flies exclusively reared from *L. phyllopus.* These markers occurred even though the host species were collected from a mixed genotype sorghum crop at a single field site in North Central Florida. While sympatric speciation accompanied by a host shift is less common in parasitic dipterans that mate away from the host, it is more likely to occur in abundant and widely distributed species like *T. pennipes* [45]. Adaptive divergence in mitochondrial DNA may disrupt the mito-nuclear coevolution of hybrids and lead to reproductive isolation and speciation [46]. Alternatively, the separate mitochondrial markers could have been introduced by parasitized *A. tristis* transported to Florida on cucurbits from the northeastern US. The extent of geneflow between the two lineages of *T. pennipes* in the field is unknown. Additional research on host preference and suitability, and mate selection is necessary to clarify the functional relationships between these lineages.

## Figures and Tables

**Figure 1 genes-14-01172-f001:**
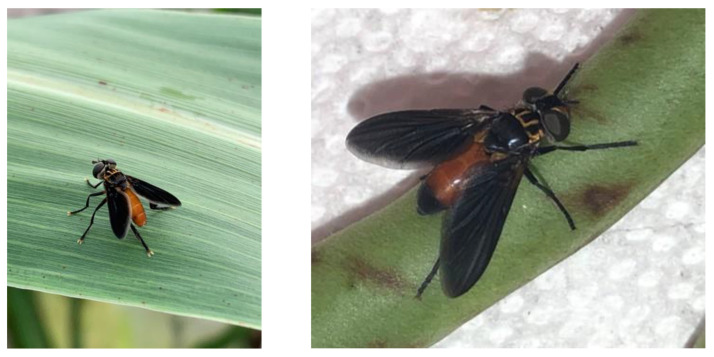
Images of *T. pennipes* collected from the study site (**left**) and reared in the laboratory (**right**).

**Figure 2 genes-14-01172-f002:**
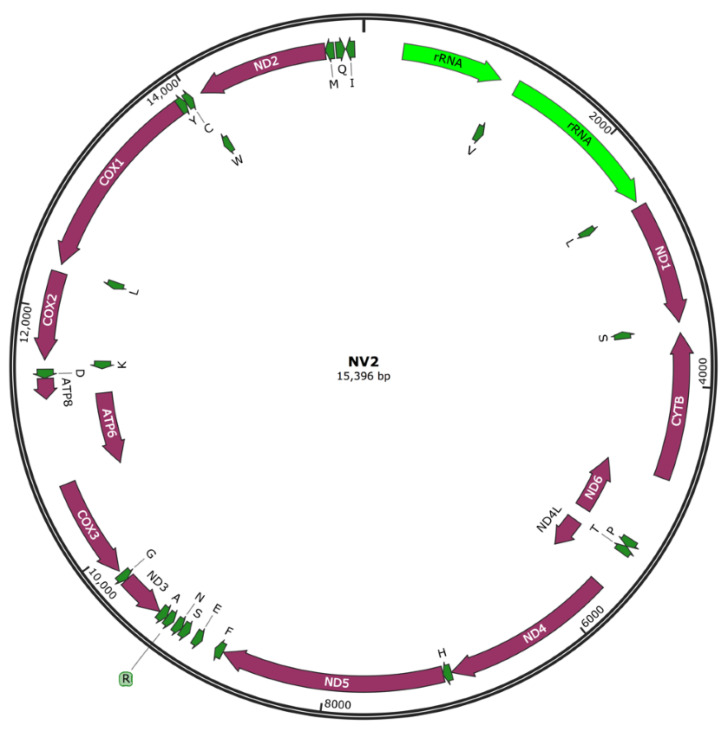
Schematic illustration of the mitochondrial genome of *T. pennipes*. PCGs: red; tRNAs: green. Outer side represents J-strand and inner side N-strand.

**Figure 3 genes-14-01172-f003:**
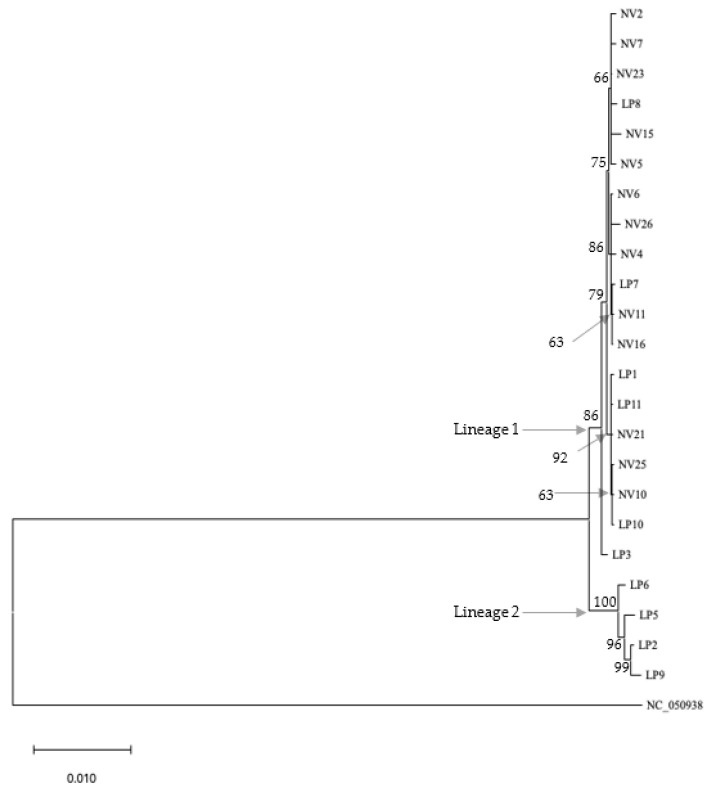
Maximum likelihood (ML) tree generated using combined sequence information from the 15 mitochondrial genes (with highest log likelihood equaling—21,231.33) from the 23 samples (13 NV and 10 LP). Bootstrap values above 50 are next to branches. Number of substitutions per site are indicated by the scale bar.

**Figure 4 genes-14-01172-f004:**
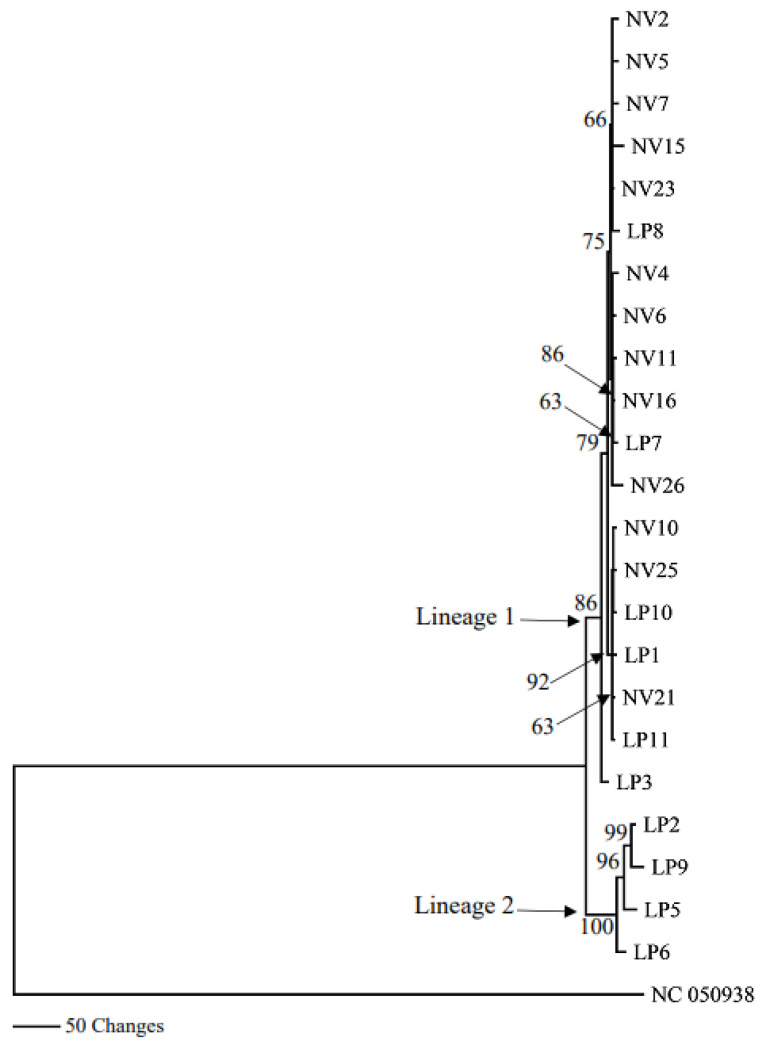
Maximum parsimony (MP) tree generated using combined sequence information from the 15 mitochondrial genes. Bootstrap values above 50 are next to branches. Scale bar indicates number of changes.

**Figure 5 genes-14-01172-f005:**
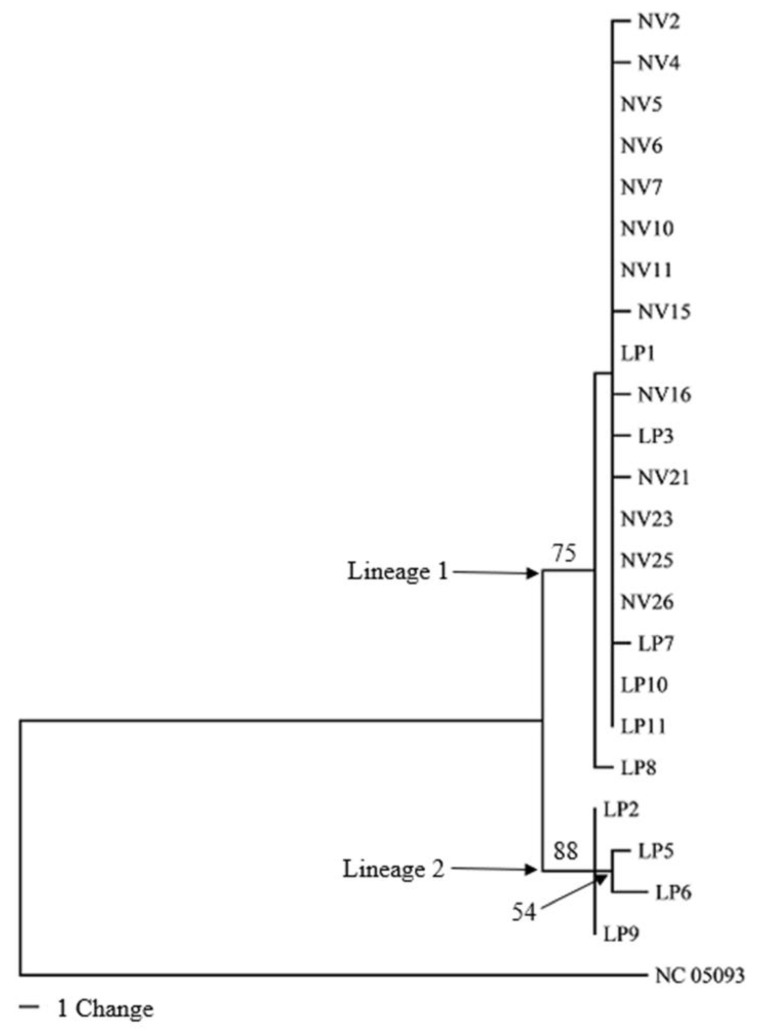
Maximum parsimony (MP) tree based on sequence information from Cox III. Bootstrap values above 50 are next to branches. Scale bar indicates number of changes.

**Figure 6 genes-14-01172-f006:**
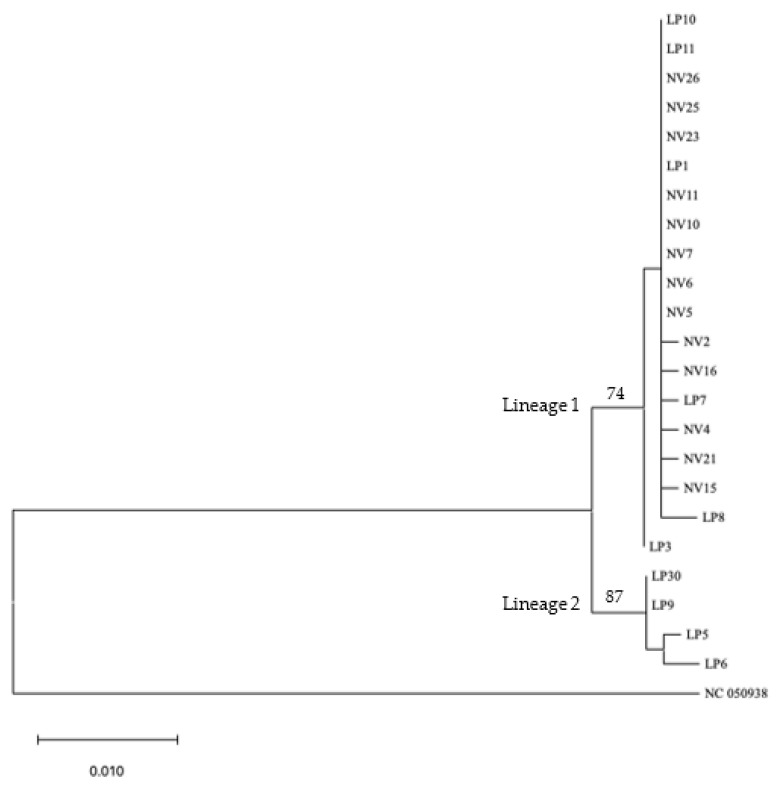
Maximum likelihood (ML) tree based on sequence information from Cox III (the highest log likelihood equaling −1423.65). Bootstrap values above 50 are next to branches. Scale bar indicates number of changes.

**Table 1 genes-14-01172-t001:** Assembly of *T. pennipes* (NV2) nuclear genome.

Assembly Statistics	
Size (MB)	672
Number of contigs Largest contigs (MB)	561 28
GC content (%)	31.7
N50 value (MB)	11.9
Number of predicted genes	1367
Complete BUSCOs	1360 (99.4%)
Complete and single-copy BUSCOs	1332 (97.4%)
Complete and duplicated BUSCOs	28 (2.0%)
Fragmented BUSCOs	3 (0.2%)
Missing BUSCOs	4 (0.4%)

**Table 2 genes-14-01172-t002:** Mitochondrial sequencing statistics for the 38 *T. pennipes* flies from *N. viridula* (NV) and *L. phyllopus* (LP). Contig sizes are for complete genomes.

Sample ID	No. Reads	No. Bases	Quality	Unannotated Reads	Contig Size (bp)	AT%
NV1	12,671	13,160,480	64	-	-	-
NV2	41,125	42,567,447	53	374	15,396	79.62
NV3	17,292	17,961,830	54	-	-	-
NV4	180,989	189,724,559	51	652	15,635	79.84
NV5	22,630	24,017,062	54	-	-	-
NV6	22,535	24,146,808	57	-	-	-
NV7	110,361	115,066,499	51	689	15,671	79.91
NV8	51,773	57,286,544	52	-	-	-
NV9	24,866	23,880,487	54	-	-	-
NV10	20,657	20,860,611	69	-	-	-
NV11	16,898	16,575,717	55	-	-	-
NV12	14,534	16,708,672	56	-	-	-
NV13	63,726	62,656,733	50	-	-	-
NV14	47,523	46,837,257	51	-	-	-
NV15	11,798	12,414,661	58	-	-	-
LP1	71,366	86,228,837	54	3426	18,399	78.52
NV16	25,677	29,132,101	54	-	-	-
LP2	67,150	74,432,675	52	-	-	-
LP3	35,033	35,413,794	55	424	15,345	79.59
NV17	26,902	26,680,575	59	-	-	-
NV18	8626	8,261,140	63	-	-	-
NV19	9404	9,392,892	57	-	-	-
NV20	53,549	54,982,744	51	-	-	-
LP4	4880	4,832,986	67	-	-	-
NV21	57,424	67,139,567	53	-	-	-
NV22	14,672	12,526,413	56	-	-	-
NV23	39,343	36,938,437	51	-	-	-
NV24	6290	4,868,262	49	-	-	-
NV25	21,492	19,843,073	53	416	15,389	80.08
NV26	51,002	57,801,828	53	-	-	-
NV27	64,355	69,748,804	52	-	-	-
LP5	91,453	11,3736,808	52	-	-	-
LP6	43,750	52,865,748	55	-	-	-
LP7	59,613	70,247,435	55	852	15,845	80.03
LP8	96,846	120,505,089	55	-	-	-
LP9	47,048	57,653,541	52	-	-	-
LP10	91,610	106,733,059	52	1167	16,140	80.20
LP11	69,224	72,032,812	54	1432	16,390	80.40

**Table 3 genes-14-01172-t003:** Mitogenomics for isolate NV2. The major and minor strands of the mitochondrial DNA are represented by letters J and N, respectively; and the direction of genes on these strands are indicated by letters F (clockwise) and R (counterclockwise).

Gene	No. Nucleotides	Strand	Direction	Start Codon	Stop Codon
srRNA	789	J	R	308	1096
lrRNA	1292	J	R	1226	2517
ND1	948	J	R	2560	3507
Cyt B	1137	J	F	3589	4725
ND6	525	N	F	4725	5249
ND4L	297	N	R	5386	5682
ND4	1338	J	R	5676	7013
ND5	1734	J	R	7080	8813
ND3	354	J	F	9395	9748
CoxIII	789	J	F	9836	10,624
ATP6	678	N	F	10,624	11,301
ATP8	162	J	F	11,295	11,456
CoxII	687	J	F	11,599	12,285
CoxI	1536	J	F	12,349	13,884
ND2	1017	J	F	14,087	15,103

## Data Availability

The datasets generated during and/or analyzed during the current study are available in the NCBI under BioProject PRJNA971909 and GeneBank accessions OR032586-OR032599.

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
