# Peer review of "First Description of the Nuclear and Mitochondrial Genomes and Associated Host Preference of Trichopoda pennipes, a Parasitoid of Nezara viridula"

_genes, 2023, doi:10.3390/genes14061172_

Round 1
Reviewer 1 Report
This article is unique in presenting the first genomic analysis (of which I am aware) of a tachinid parasitoid of an important agricultural pest. Tachinids are potentially important as biocontrol agents, therefore this paper will be a useful contribution. The genomic analysis supports previous suppositions based on limited behavioral data and observations that suggested evidence of cryptic species, which is valuable in considering the specificity and host range for biocontrol. I am not qualified to judge the details of the molecular biology methodology and analytical software employed, but the entomological aspects of the study appear sound, and the conclusions also. The paper is well organized and clearly written and needs little editing. I have noted a few relatively minor items by line number below.
Comments by line number:
28 species names in keywords should be italicized per usual practice for latin names
56 “but” should be “bug” in this line
98-101 The DNA extracted from these insects was intended for the mitochondrial sequencing – this should be mentioned here, because the very next sentence says DNA was extracted from one parasitoid. This is potentially confusing as written.
105 insert “the” before “DNA sample”. Is this sample the one shown in tables and figure 1 “NV2”? If so, indicate this here or somewhere appropriate early in the results.
130 insert “the” before “draft genome”
Table 2 “(bp)” is duplicated in the table header row – is there a reason for this? Also in the header row, “ases” appears below the column heading “No. bases”. Is this a typo error?
332 (14) (pdf): what is this? Should there be an author name here?
338 this citation appears incomplete: what is the source?
342-343 Why is this entire entry capitalized?
346-347 spacing needs to be adjusted between words in this citation.
Generally throughout References: Latin names are typically italicized but are not here – is this an error or is it journal formatting?
Reviewer 2 Report
The authors sequenced 38 samples of Trichopoda pennipes from different hosts, assembled their nuclear and mitochondrial genomes, and made phylogenetic analyses via ML and MP, along with a discussion of relationships between mitochondrial genotypic and hosts, resulted in that Trichopoda pennipes partially parasitized Nezara viridula and Leptoglossus phyllopus, and partially parasitized Leptoglossus phyllopus. Obviously the authors have done a lot of work and make contribution to the knowledge to Trichopoda pennipes.
1. the authors sequenced 38 samples in total, but only 9 complete mitochondrial genomes obtained, why? What’s the problem? The authors must clarify it in the manuscript.
2. the sample picture(s) (photos of Trichopoda pennipes) is missing, it is suggested that the authors should add it or them in the ms.
3. Figure 3 and figure 4 are of low resolution than others.
4. in INTRODUCTION, the authors uses a lot of words to state the use of Trichopoda pennipes to control Nezara viridula, but in RESULTS and DISCUSSION, it seems that showing weakly explanations to how to control Nezara viridula? or its mechanism is weakly clear. I suggest related sentences should be re-write.
5. the gene data is not opened in any public web seemingly, without accession number.
Round 2
Reviewer 2 Report
It seems the authors has fixed and revised the manuscript and modified related sentences accordingly, I think it is now could be accepted for publication after obtaining the numbers for the nuclear genomes and mitochondrial genomes in GeneBank.
Author Response
Thank you for your reviews. We have now included the accessions for the nuclear genome and mitogenomes in our revised manuscript.